# SciLitBench: Benchmark and Design Principles for LLM-Powered Systematic Literature Reviews

## Abstract

Systematic literature reviews are essential for science but remain labor-intensive. To benchmark and improve automation, we introduce SciLitBench, a new dataset of 42,980 curated abstracts, 2,311 full texts, and TODOXXX structured data elements (e.g. study-level metadata, PICO entities, outcome measures, and evidence) annotated and labeled for inclusion decisions and knowledge reasoning. Across 22 open-source large language models (LLMs), we uncover general design principles that make automation reliable under recall-skewed objectives ($F_2$). First, we observe clear scaling and prompt-design effects: explicit inclusion/exclusion prompting improves accuracy by up to +29%, while adding researcher "thought traces" yields a +28% gain. Second, we show that reliability under full-text screening depends sharply on the interaction between context length and model capacity. Motivated by this, we introduce a token-length–aware routing system that surpasses ensembles of strongest models ($F_2 = 0.949$ vs $0.938$). Finally, we demonstrate a human-in-the-loop, rubric-guided extraction workflow that separate field extraction from guideline adherence checking to align model outputs with domain standards given researcher feedback. Together, our benchmark and findings establish scaling, prompt design, thought traces, and adaptive routing as key principles for reliable, researcher-aligned automation of systematic reviews.

## 1 Introduction

Systematic literature reviews are central to scientific process: they synthesize evidence across studies, establish consensus on contested topics, clarify conceptual boundaries, and chart promising directions for future research. Meta-analyses of medical interventions, for instance, have shaped public health policy on smoking, nutrition, and preventive care, underscoring how reviews can accelerate scientific progress and improve societal outcomes (Hackshaw et al., 1997; Mozaffarian et al., 2006; Cholesterol Treatment Trialists' (CTT) Collaboration et al., 2010; Ntais & Talias, 2024; Office of the Surgeon General (US) & Office on Smoking and Health (US), 2004). The demand for such syntheses has grown rapidly: despite a slow and labor-intensive effort, the number of published literature reviews has risen dramatically in recent years, reflecting both their impact and the need for tools that can keep pace with the expanding scientific record beyond human curation capacity (Hoffmann et al., 2021; Smela et al., 2023).

**Why automation, and why now?** Conducting a high-quality review involves multi-stage decision-making – search and retrieval, title/abstract screening, full-text screening, and structured data extraction – each demanding careful, consistent judgment across thousands of records. The resulting time and cost burdens create real bottlenecks for science. Meanwhile, AI capabilities are evolving quickly, inviting us to modernize review pipelines without compromising rigor.

The recent rise of LLMs has transformed the landscape of AI research and captured the public imagination. These models are trained on a vast corpora of text and exhibit emergent capabilities that go beyond simple pattern recognition. Since the release of ChatGPT based on GPT-3.5, progress has been remarkably fast in a wide variety of fronts (Maslej et al., 2025; Ho et al., 2024). New generations of reasoning models, such as o1 and other state-of-the-art systems (OpenAI et al., 2024; DeepSeek-AI et al., 2025), now excel on benchmarks requiring multi-step structured problem-solving, and even achieved Olympic gold medals in math and programming (OpenAI et al., 2025).

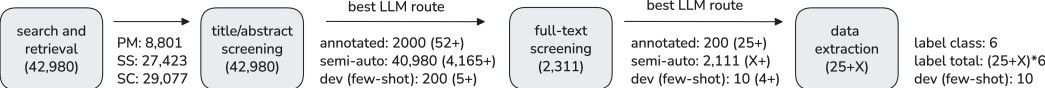

Figure 1: **SciLitBench.** The dataset is curated on the topic of the use of AI to assist literature reviews. With a librarian-optimized search query, we retrieved 42,980 peer-reviewed publication records from three databases: PubMed (PM), Semantic Scholar (SS), and Scopus (SC). The "+" indicates the number of positive labels.

As such, LLMs are naturally well-suited for the task of literature review automation. They encode extensive background knowledge and have been exposed to scientific literature during pretraining; they are naturally adept at working with textual data and, with the steady increase of context window sizes, are now capable of processing entire papers in a single pass. At the same time, the rate of factual errors and hallucinations has decreased in the strongest models, suggesting that these systems can be utilized to reliably automate the tasks involved in literature reviews (Muhlgay et al., 2024; Wei et al., 2024a;b). Harnessing LLMs for review automation therefore represents a critical step toward ensuring that the pace of evidence synthesis can match the pace of discovery.

**What's missing in the current landscape?** Prior academic work typically tackles a single stage (e.g., title/abstract screening, full-text screening, or extraction), often on small datasets with limited evaluation scope. Recent studies probe prompting or ensembling but still lack breadth and unified protocols. Complementing these are commercial tools (e.g., general "AI review assistants") whose methods and evaluations are opaque. Together, these leave open crucial questions: *Which design choices actually make LLMs reliable collaborators? How do capacity and context length interact? What evaluation protocols should the community standardize on?*

**Our approach.** We introduce *SciLitBench*, a benchmark suite and open artifact release aimed at both evaluating and improving LLM-powered automation across the review pipeline. Starting from 42,980 retrieved records, we annotate 2,000 titles/abstracts and 200 full texts with inclusion labels and reasoning/rationales. We evaluate 22 open-source LLMs across three key stages – title/abstract screening, full-text screening, and data extraction – while deriving general design principles (e.g., explicit inclusion/exclusion prompting, scaling, few-shot learning, chain-of-thought, thought traces, and length-aware routing) for making them reliable collaborators. In abstract and full-text screening we emphasize recall with F2 (while also reporting precision/recall), reflecting the real-world priority of minimizing false exclusions in early stages. We carry best-performing systems forward to full-text screening and then to extraction, quantifying workload savings and reliability throughout. We then semi-automate the label curation for the full 43k publication records and 2,311 available full-text PDFs to create 45,291 binary labels and TODOXXX multi-class labels as a multi-task dataset and benchmark (Fig. 1. We also release codes, prompts, outputs, scoring scripts, and guidelines to maximize reproducibility and community uptake. We have four **key contributions:**

1. A multi-stage, open benchmark for review automation (SciLitBench), with frozen splits and documented protocols spanning abstract screening, full-text decisions, and extraction.

2. Comprehensive evaluation of 22 LLMs with standardized prompts and metrics, revealing robust gains from explicit inclusion/exclusion prompting and researcher thought traces.

3. Reliability test at long context that shows a performance interaction between model capacity and document length, motivating a token-length–aware routing strategy that surpasses strong static ensembles under recall-centric objectives.

4. Human-in-the-loop extraction workflow with a rubric-guided collaborator/checker pattern to align outputs with researcher standards and operationalize trustworthy principles.

Unlike proprietary tools, SciLitBench prioritizes open methods and verifiable evaluation. Our approach prioritizes transparency: frozen splits, versioned prompts, outputs, and evaluation scripts are released to enable direct replication and leaderboard-style comparisons. We explicitly report recall-skewed screening ($F_2$), strict format adherence for structured outputs, and workload/cost curves, providing LLM-centric diagnostics that complement existing resources and proprietary systems while anchoring automation research in open, verifiable science. By systematically documenting where models succeed and fail, we aim to ground practical adoption in shared, inspectable evidence and provide a stable foundation for iterative improvement by the community.

| Benchmark | Domain | Stages | Long | Rats. | Task | Size |
|---|---|---|---|---|---|---|
| **SciLitBench (ours)** | SR | TA+FT+Ext | ✓ | ✓ | Bin + Struct | 43k TA; 2k FT |
| SYNERGY De Bruin et al. (2023) | SR | TA | ✗ | ✗ | Bin | 169k recs / 26 SRs |
| Evidence Inference DeYoung et al. (2020) | RCTs | FT (effect) | ✓ | ✓ | Cls (ICO) | 7k query–doc inst. |
| EBM-NLP (PICO) (Nye et al., 2018) | RCT abs. | Ext (P/I/O) | ✗ | ( spans ) | NER | 5k abs. |
| $MS^2$ (DeYoung et al., 2021) | Med SRs | Summ. (MD) | ✓ | ✗ | Summ. | 470k docs; 20k sums |
| BioASQ (Tsatsaronis et al., 2015) | Biomed IR/QA | IR+QA | ( snips ) | ✓ | IR+QA | Yearly batches |
| PubMedQA (Jin et al., 2019) | Biomed abs. | QA (YNM) | ✗ | ✗ | QA | 1k labels / 211k auto |

Table 1: **SciLitBench in context.** SciLitBench uniquely spans *three* SR stages on long PDFs with inclusion rationales and an LLM-centric protocol (recall-skewed $F_2$, strict format adherence, and length-aware routing). (SR: systematic reviews; TA: title/abstract screening; FT: full-text screening/decision; Ext: structured extraction; Long: long documents (full-text PDFs); Rats.: released human rationales; Bin: binary include/exclude; Struct: structured fields; ICO: Intervention versus a Comparator for a specific Outcome; Summ.: summarization; MD: multi-document; YNM: yes/no/maybe; RCT: random controlled trials; IR: information retrieval.)

## 2 RELATED WORK

**AI and LLMs for literature reviews.** A growing body of academic work has explored how AI methods can automate parts of the literature review pipeline. Studies often focus on singular tasks such as title/abstract screening, full-text screening, or data extraction (Tran et al., 2023; Jensen et al., 2014; Li & Kanoulas, 2019; Przybyła et al., 2018; Li et al., 2024; Gao & Zhang, 2015; Ferracane et al., 2016). More recent work tests LLMs for prompt tuning, ensembling, or similar design choices (Homiar et al., 2025; Igoli et al., 2024; Kılıç et al., 2024; Huotala et al., 2024; Iacus et al., 2025; Sanghera et al., 2024), but typically with small datasets and limited evaluation scope. These LLM studies most remain single-stage case studies on abstract-only corpora, often without rationale release, long-PDF inputs, recall-skewed metrics ($F_2$), strict output-format checks, or workload/cost reporting. As context windows expand, capacity–length interactions and format adherence emerge as primary failure modes; few benchmarks explicitly analyze these factors or standardize evaluation around document length and structured-output contracts.

Complementary to these experimental studies, survey papers summarize the role of LLMs in literature review automation (Scherbakov et al., 2025; Zhuang et al., 2025; Galli et al., 2025). However, many reviews have been conducted without automation at all, and those using AI rarely analyze the methodological details of what designs succeed or fail. As a result, these papers provide broad overviews but not concrete insights into the design principles necessary for reliable automation.

**Datasets and benchmarks.** Several widely used datasets target *subtasks* relevant to evidence synthesis rather than the end-to-end review workflow. SYNERGY provides high-quality include/exclude labels for *title/abstract screening* across multiple topics, enabling screening simulations but not full-text judgments or extraction (De Bruin et al., 2023). EBM-NLP frames *structured extraction* as PICO span labeling from random controlled trials (RCT) abstracts (Population, Intervention, Outcome) (Nye et al., 2018). Evidence Inference evaluates *full-text effect direction* (increase/decrease/no-effect) with evidence sentences, but not systematic review (SR) inclusion decisions or schema-level extraction (DeYoung et al., 2020). $MS^2$ focuses on *multi-document summarization* linking SRs to constituent trials (DeYoung et al., 2021). Biomedical IR/QA resources such as BioASQ and PubMedQA benchmark retrieval, snippet-grounded QA, and abstract-level decision QA, respectively (Tsatsaronis et al., 2015; Jin et al., 2019). These resources are invaluable, yet they do not jointly evaluate *screening → full-text decision → structured extraction* under long-document inputs or LLM-specific reliability criteria. Table 1 summarizes these contrasts.

**Positioning of *SciLitBench*.** *SciLitBench* addresses these limitations by providing, to our knowledge, the largest curated public dataset and protocol suite that *spans three stages* of the SR pipeline on *long-document* inputs: starting from 42,980 retrieved records, we release annotated titles/abstracts and full texts with inclusion labels and *human-readable rationales*. We conduct a comprehensive evaluation of 22 open-source LLMs across title/abstract screening, full-text screening, and schema-guided extraction. Beyond metrics, we distill design principles: explicit inclusion/exclusion prompting, researcher thought-traces, and *length-aware routing* that captures capacity–length interactions–not systematically studied in prior work. For extraction, where open-endedness complicates evaluation, we propose a collaborator–checker workflow: (i) co-design guidelines with an LLM and the human researcher; (ii) separate extraction from guideline-adherence checking, improving reliability and aligning outputs with researcher preferences.

# 3 SciLitBench: Tasks, Data, and Evaluation Protocol

Here we specify SciLitBench as a reusable protocol: task schemas and I/O contracts (Sec. 3.1), dataset curation and frozen splits (Sec. 3.2), evaluation metrics (Sec. 3.3), baseline LLMs and inference settings (Sec. 3.4). Later sections instantiate the protocol per stage with results and ablations.

## 3.1 Tasks and Input/Output Contracts

**Notation.** Let $\mathcal{D}$ be the corpus of candidate records returned from a predefined search strategy. For document $d \in \mathcal{D}$, let $t_d$ be the title, $a_d$ the abstract, and $f_d$ the full text (potentially a long PDF). Let $y_d \in \{\texttt{include}, \texttt{exclude}\}$ be the gold screening label and $r_d$ a human-provided rationale (quoted evidence + brief note).

**Stage A – Title/Abstract screening (TA).** *Input:* $(t_d, a_d)$. *Output:* $\hat{y}_d \in \{\texttt{include}, \texttt{exclude}\}$ and optional $\hat{r}_d$. *Contract:* single binary decision; if rationales are requested, $\hat{r}_d$ must contain a short quote from $a_d$ and a one-sentence reason.

**Stage B – Full-text screening (FT).** *Input:* $f_d$ (long-document). *Output:* $\hat{y}_d$ and optional $\hat{r}_d$. *Contract:* same decision space as TA; when rationales are requested, $\hat{r}_d$ must cite page/section anchors or verbatim spans.

**Stage C – Structured extraction (Ext).** *Input:* $f_d$ for documents with $y_d = \texttt{include}$. *Output:* a JSON object conforming to a public schema $\mathcal{S}$ (e.g., fields for Population/Intervention/Comparator/Outcome, study design, $n$, effect size, etc.). *Contract:* strict schema adherence (all required keys present; values meeting type/format constraints such as numeric formats and controlled vocabularies).

**Format-adherence (*Reliability*) checks.** For stages requesting rationales or structured outputs, we enforce a deterministic validator $\mathcal{V}$ (regex + JSON schema). For stages requesting binary outputs, we enforce $\mathcal{V}$ (regex). Predictions failing $\mathcal{V}$ are counted as *invalid* and scored as errors.

## 3.2 Dataset and frozen splits

We begin from 42,980 retrieved records using a fixed multi-database search using a librarian-optimized query (Appx. A). After de-duplication, eligibility screening, and full text retrieval, we annotate and publish *frozen* splits with stable identifiers:[1]

**TA set:** 2,000 title/abstract items with $y_d$ and $r_d$ (quote+reason), with splits `dev/train/test`: 100/100/1800. The best performing system is used to automatically label the rest 40,980 title/abstract items with sample validation. The `dev` set is used for few-shot examples.

**FT set:** 200 full texts with $y_d$ and $r_d$ (evidence spans + notes), with splits `dev/train/test`: 10/100/90. The best performing system is used to automatically label the rest 2,111 full-text items with sample validation. The `dev` set is used for few-shot examples.

## 3.3 Evaluation metrics

**Screening (TA/FT).** Our primary metric is $F_2$, the recall-skewed $F_\beta = (1 + \beta^2) \frac{\text{Precision} \cdot \text{Recall}}{\beta^2 \cdot \text{Precision} + \text{Recall}}$ with $\beta = 2$. We also report Precision, Recall, and confusion matrices. Because early-stage false negatives are costly, $F_2$ is the model-selection metric for screening.

**Reliability Rate (format adherence).** For any stage with output constraints, Rel. Rate $= 1 - \frac{\#\{\text{predictions failing } \mathcal{V}\}}{\#\{\text{predictions}\}}$. Invalid predictions are scored as errors for task metrics (binary: treated as wrong class; extraction: zero credit for affected fields).

**Structured extraction.** We evaluate at three granularities: (i) *Exact-match* per field, (ii) *Schema-match* allowing benign normalization (e.g., whitespace, unit aliases), and (iii) *Record completeness* (fraction of required fields filled and valid).

---

[1] We release DOIs/PMIDs and hash-based IDs; PDFs are redistributed only where licensing permits.

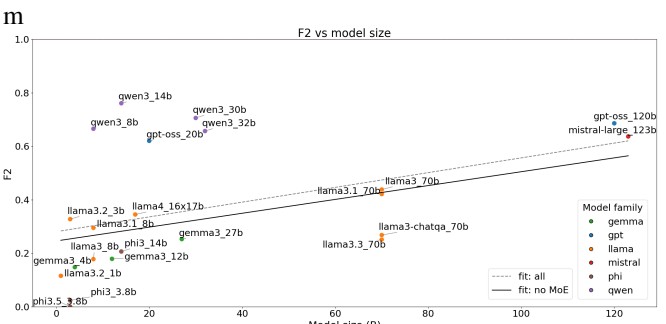

Figure 2: **Scaling of title/abstract screening performance by model size.** $F_2$ scores are shown for 22 open-source LLMs across families and parameter scales. A clear scaling law emerges, with smaller models ($<20B$) generally performing poorly, while larger models achieve substantially higher reliability. Notably, reasoning-oriented variants (e.g., Qwen3, gpt-oss) attain strong performance despite moderate size.

| model | model size | zero-shot | | | few-shot 2 | | | few-shot 5 | | |
|---|---|---|---|---|---|---|---|---|---|---|
| | | basic | incl | incl+excl | basic | incl | incl+excl | basic | incl | incl+excl |
| Gemma3 | 12b | 0.179 | 0.212 | 0.257 | 0.188 | 0.219 | 0.317 | 0.329 | 0.307 | 0.339 |
| Gemma3 | 27b | 0.253 | 0.293 | 0.421 | 0.227 | 0.249 | 0.324 | 0.359 | 0.330 | 0.428 |
| Gemma3 | 4b | 0.148 | 0.186 | 0.157 | 0.129 | 0.136 | 0.146 | 0.154 | 0.151 | 0.164 |
| gpt-oss | 120b | 0.686 | 0.681 | 0.697 | 0.687 | 0.684 | 0.675 | 0.674 | 0.659 | 0.686 |
| gpt-oss | 20b | 0.620 | 0.655 | 0.628 | 0.644 | 0.658 | 0.679 | 0.677 | 0.694 | 0.686 |
| Llama3 | 70b | 0.268 | 0.213 | 0.287 | 0.322 | 0.390 | 0.472 | 0.315 | 0.410 | 0.466 |
| Llama3.1 | 70b | 0.421 | 0.598 | 0.755 | 0.395 | 0.577 | $0.751^{+}$ | 0.302 | 0.531 | $0.749^{++}$ |
| Llama3.1 | 8b | 0.295 | 0.535 | 0.609 | 0.210 | 0.456 | 0.646 | 0.459 | 0.534 | 0.588 |
| Llama3.2 | 1b | 0.115 | 0.195 | 0.186 | 0.174 | 0.181 | 0.197 | 0.128 | 0.132 | 0.148 |
| Llama3.2 | 3b | 0.327 | 0.579 | 0.564 | 0.292 | 0.335 | 0.410 | 0.318 | 0.299 | 0.430 |
| Llama3.3 | 70b | 0.251 | 0.497 | $0.786^{*}$ | 0.238 | 0.384 | $0.717^{+++}$ | 0.228 | 0.464 | $0.735^{+}$ |
| Llama3 | 70b | 0.438 | 0.567 | 0.679 | 0.283 | 0.358 | 0.570 | 0.288 | 0.471 | 0.698 |
| Llama3 | 8b | 0.178 | 0.360 | 0.452 | 0.205 | 0.312 | 0.476 | 0.310 | 0.377 | 0.542 |
| Llama4 | 16x17b | 0.345 | 0.478 | 0.482 | 0.431 | 0.620 | 0.737 | 0.556 | 0.625 | 0.669 |
| Mistral | 123b | 0.637 | 0.610 | 0.606 | $0.766^{*}$ | 0.671 | 0.570 | 0.728 | 0.638 | 0.614 |
| Phi3.5 | 3.8b | 0.000 | 0.000 | 0.000 | 0.000 | 0.000 | 0.000 | 0.065 | 0.043 | 0.041 |
| Phi3 | 14b | 0.206 | 0.217 | 0.246 | 0.179 | 0.355 | 0.477 | 0.207 | 0.271 | 0.295 |
| Phi3 | 3.8b | 0.023 | 0.076 | 0.077 | 0.147 | 0.164 | 0.245 | 0.192 | 0.217 | 0.258 |
| Qwen3 | 14b | $0.761^{*}$ | 0.720 | 0.749 | $0.739^{+}$ | 0.753 | 0.736 | 0.749 | $0.756^{*}$ | 0.736 |
| Qwen3 | 30b | 0.706 | 0.567 | 0.565 | $0.726^{+}$ | $0.760^{*}$ | 0.712 | 0.734 | 0.734 | 0.751 |
| Qwen3 | 32b | 0.657 | 0.682 | 0.668 | 0.754 | 0.704 | 0.728 | 0.706 | 0.726 | 0.740 |
| Qwen3 | 8b | 0.666 | 0.646 | 0.665 | 0.686 | 0.660 | 0.667 | 0.638 | 0.665 | 0.725 |

| | |
|---|---|
| strong pool ($t_s$=3) | **0.808** (precision: **0.551**, recall: 0.915) |
| lenient pool ($t_l$=9) | 0.803 (precision: 0.512, recall: 0.936) |
| comb. pool ($t_s = 3, t_l = 4$) | 0.655 (precision: 0.275, recall: **1.000**) |

Table 2: **Systematic evaluation in title/abstract screening.** We report $F_2$ (recall-weighted $F$-score) for each model across prompting and few-shot configurations. From single-model results, we select the top five configurations to form a *strong pool* (marked $^{*}$) and build a majority-vote ensemble (predict "include" if at least $t_s$ of 5 vote yes). To prioritize high recall, we also form a *lenient pool* (marked $^{+}$) of the ten best configurations with recall $= 1.0$ and test majority-vote thresholds $t_l$. We further evaluate a *combined* ensemble that predicts "include" if either pool crosses its threshold. For these three ensemble variants we report $F_2$, precision, and recall, and select the best system subject to recall $= 1.0$ for curation.

## 3.4 Baseline LLMs and Inference Settings

We standardize two families: **1) Open LLMs**: 22 open-source models spanning small/medium/large capacities and varying context windows and parameter scales: Llama (Touvron et al., 2023), GPT-OSS (Agarwal et al., 2025), Gemma (Team et al., 2024), Qwen (Bai et al., 2023), Mistral (Jiang et al., 2024), and Phi (Abdin et al., 2024). Decoding is standardized (temp, top-$p$). Each model is queried in zero-shot and few-shot conditions under different prompting strategies. Performance is measured on the held-out evaluation set. Prompts are versioned and frozen per stage. and **2) Ensembles/Routing**: We include (i) best single model, (ii) static majority or weighted score ensembles, and (iii) a token-length–aware router that adapt ensemble weighting given the document.

Unless stated otherwise: temperature $= 0$, top-$p$=0.9, max tokens set to avoid truncation of required outputs, with retries on format failure (max 1 retry with stricter system prompt). For long documents ($f_d$), we pass full text up to the model's context limit. Full prompts are included in Appx. B.

## 4 Stage A – Title/Abstract Screening (TA)

**Problem definition and inputs.** Given $(t_d, a_d)$ for each record $d$, the model outputs a binary decision $\hat{y}_d \in \{\texttt{include}, \texttt{exclude}\}$ and, when requested, a short rationale $\hat{r}_d$ (quote + one-sentence reason). Evaluation uses the `TA dev/train/test` splits in Sec. 3.2. Metrics and validators

| model | model size | zero-shot | | | | few-shot | | | |
| | | one-step | | two-step | | no thought traces | | thought traces | |
| | | basic | strict | unstruct. | CoT | few-shot 2 | few-shot 4 | few-shot 2 | few-shot 4 |
|---|---|---|---|---|---|---|---|---|---|
| Gemma3 | 12b | 0.513 | 0.611 | $0.694^+$ | $0.758^+$ | 0.580 | 0.568 | 0.636 | 0.565 |
| Gemma3 | 27b | 0.670 | 0.708 | 0.660 | 0.758 | 0.619 | 0.515 | 0.670 | 0.606 |
| Gemma3 | 4b | 0.180 | 0.476 | 0.528 | $0.595^+$ | 0.586 | 0.636 | 0.524 | 0.524 |
| gpt-oss | 120b | 0.684 | 0.729 | 0.123 | 0.040 | 0.038 | 0.000 | 0.258 | 0.312 |
| gpt-oss | 20b | $0.852^*$ | $0.862^*$ | 0.824 | $0.882^*$ | 0.677 | 0.684 | 0.824 | $0.872^*$ |
| Llama3 | 70b | 0.037 | 0.000 | 0.100 | 0.189 | 0.144 | 0.198 | 0.097 | 0.053 |
| Llama3.1 | 70b | 0.676 | 0.664 | $0.743^+$ | $0.758^+$ | 0.441 | 0.000 | 0.660 | 0.698 |
| Llama3.1 | 8b | 0.688 | 0.636 | 0.673 | $0.658^+$ | 0.583 | 0.000 | 0.625 | 0.538 |
| Llama3.2 | 1b | 0.000 | 0.385 | 0.330 | 0.230 | 0.110 | 0.000 | 0.357 | 0.537 |
| Llama3.2 | 3b | 0.491 | 0.496 | $0.676^+$ | 0.652 | 0.602 | 0.458 | 0.512 | 0.507 |
| Llama3.3 | 70b | 0.625 | 0.591 | 0.676 | 0.765 | $0.664^{++}$ | 0.654 | 0.688 | 0.699 |
| Llama3 | 70b | 0.280 | 0.244 | 0.673 | 0.693 | 0.700 | 0.700 | 0.700 | 0.677 |
| Llama3 | 8b | 0.195 | 0.296 | 0.532 | 0.474 | 0.612 | 0.612 | 0.571 | 0.561 |
| Llama4 | 16x17b | 0.038 | 0.081 | 0.294 | 0.424 | 0.361 | 0.353 | 0.361 | 0.412 |
| Mistral | 123b | 0.533 | 0.533 | 0.735 | 0.526 | - | - | - | - |
| Phi3.5 | 3.8b | 0.000 | 0.000 | 0.204 | 0.248 | 0.323 | 0.323 | 0.269 | 0.266 |
| Phi3 | 14b | 0.091 | 0.465 | 0.556 | 0.747 | 0.517 | 0.503 | 0.517 | 0.551 |
| Phi3 | 3.8b | 0.236 | 0.128 | 0.152 | 0.449 | 0.510 | 0.510 | 0.433 | 0.462 |
| Qwen3 | 14b | 0.714 | 0.815 | 0.781 | 0.798 | 0.305 | 0.349 | 0.292 | 0.292 |
| Qwen3 | 30b | 0.698 | 0.595 | 0.739 | 0.739 | 0.181 | 0.076 | 0.152 | 0.152 |
| Qwen3 | 32b | 0.824 | 0.824 | 0.798 | 0.833 | 0.539 | 0.426 | 0.395 | 0.491 |
| Qwen3 | 8b | $0.862^*$ | 0.815 | 0.814 | $0.781^+$ | 0.333 | 0.285 | 0.268 | 0.254 |

| | |
|---|---|
| strong pool ($t_s$=5) | 0.938 (precision: 0.750, recall: **1.000**) |
| lenient pool ($t_l$=10) | 0.904 (precision: 0.652, recall: **1.000**) |
| comb. pool ($t_s = 5, t_l = 10$) | 0.882 (precision: 0.600, recall: **1.000**) |
| routing ensemble | **0.949** (precision: **0.789**, recall: **1.000**) |

Table 3: **Systematic evaluation in full-text screening.** We report $F_2$ (recall-weighted $F$-score) for each model across prompting and few-shot configurations. From single-model results, we select the top five configurations to form a *strong pool* (marked $^*$) and build a majority-vote ensemble (predict "include" if at least $t_s$ of 5 vote yes). To prioritize high recall, we also form a *lenient pool* (marked $^+$) of the ten best configurations with recall = 1.0 and test majority-vote thresholds $t_l$. We further evaluate a *combined* ensemble that predicts "include" if either pool crosses its threshold, and a *routing* ensemble that fits a logistic regression over model weighting with bins by document context length. For these four ensemble variants we report $F_2$, precision, and recall, and select the best system subject to recall = 1.0 for curation.

follow Sec. 3.3; the primary selection metric is $F_2$, which emphasizes recall over precision, though we also track precision and recall explicitly in downstream voting systems.

In this stage we evaluate whether LLMs can reliably automate the inclusion decisions of systematic reviews from titles and abstracts alone. We include a wide variety of model families and sizes and investigate different prompts, few-shot, and voting mechanisms with the aim of finding the best performance. Our selected system achieves perfect recall in our dataset while also significantly reducing workload. Finally, this system is applied to classify the unlabeled list of papers form our initial search, which will conform the papers selected for the next stage of the review, full-text screening. The overall aim in this stage is twofold: (1) explore the design factors that influence performance under abstract screening, and (2) select the best-performing system to apply at scale to the remaining abstracts with sample validation, serving as a filtered pool for full-text screening. We release the full 42,980 original title/abstracts, their semi-automated labels and researcher-annotated rationales, as our first contribution to SciLitBench.

**Scaling law.** Table 2 reports $F_2$ across all model configurations. A clear scaling trend emerges: larger models consistently achieve higher $F_2$ scores (Fig. 2). While mixture-of-experts and reasoning-oriented variants tend to perform well even at moderate sizes, the overall regression line shows a strong positive relationship between parameter count and reliability. This provides direct evidence that model size is an important contributor to abstract screening performance.

**Prompt tuning.** We design three prompt variants to test the effect of instruction-following: (i) basic, a generic classification query; (ii) inclusion, which explicitly asks whether the abstract meets inclusion criteria; and (iii) inclusion+exclusion, which additionally requires explicit exclusion reasoning (prompts provided in Appx. B.1. Results show substantial gains over the basic baseline: inclusion prompting yields a +17.0% relative improvement, and inclusion+exclusion prompting yields +28.8% (Fig. 3a). This establishes prompt design as a simple but powerful lever for improving performance in review automation.

**Few-shot learning.** We next evaluate whether few-shot demonstrations improve performance, varying both the number of examples (2, 5) and the proportion of positive cases (0.0, 0.5, 1.0). On

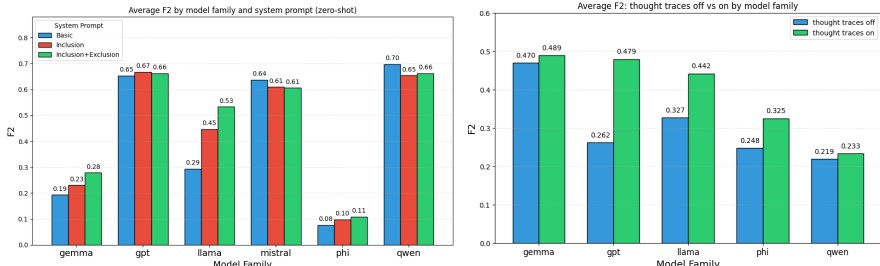

Figure 3: **Effect of system prompt tuning on title/abstract screening (zero-shot) and full-text screening (few-shot.** (Left panel, a) In TA, compared to a basic classification query, prompts that explicitly reference inclusion criteria improve performance (+17.0%) in average, while prompts requiring both inclusion and exclusion reasoning yield the largest gain (+28.8%). This highlights prompt design as a key driver of reliability in review automation. (Right panel, b) In FT, incorporating rationales (i.e. researcher "thought traces") into few-shot examples improves conformity with sophisticated criteria, raising average $F_2$ from 0.309 to 0.396 (+28%). This shows that exposing models to both decisions and the reasoning behind them is a powerful lever for improving reliability in review automation.

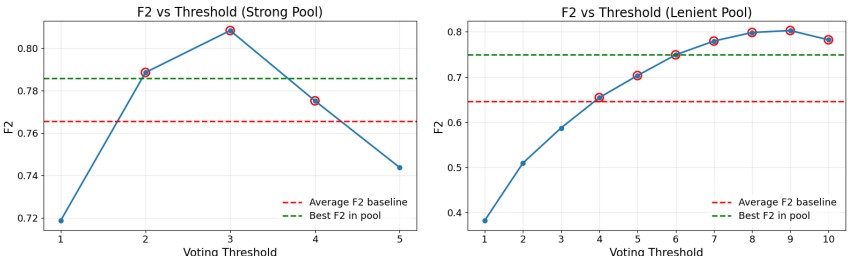

Figure 4: **Voting mechanisms improves performance.** Among both the strong pool (left panel) and lenient pool (right panel), the $F_2$ follows an inverted-U pattern: at low thresholds ($t_s = 1$, $t_l = 1$) nearly all candidate papers are included, while at high thresholds ($t_s = 5$, $t_l = 10$) the voting becomes too restrictive. A balanced threshold of $t_s = 3$ and $t_l = 9$ achieves the best trade-off, yielding $F_2 = 0.808$ and $0.803$, respectively, higher than both their respective baselines (0.766 and 0.646) and the best single models (0.786 and 0.749).

aggregate, few-shot performance does not exceed the zero-shot baseline (see Appx. C). Nonetheless, this experiment proves valuable: several of the later-selected best performing models achieves their strongest results in few-shot settings, indicating that while not consistently helpful, examples can be critical for extracting peak performance from certain models.

**Voting mechanisms.** Finally, we investigate whether ensembling models through majority voting can improve screening. We define two pools: a strong pool of the top five models ranked by $F_2$, and a lenient pool of the top ten models ranked by recall (with minimum precision of 0.2). For each pool, we vary the threshold required for a positive decision (which we refer to as $t_{\text{strong}}$ and $t_{\text{lenient}}$ respectively). Results reveal distinct patterns (Figures 4). In the strong pool, performance follows an inverted-U curve: thresholds of 1 or 5 lead to poor trade-offs, but a threshold of 3 maximizes $F_2$ at 0.808, outperforming both the average baseline (0.766) and the single best model (0.786). In the lenient pool, performance steadily increases with threshold, peaking at $F_2 = 0.803$ at threshold 9, again above both the baseline (0.646) and best single model (0.749).

For downstream application to the 40,980-paper test set, we select a combined configuration of $t_s = 3$, $t_l = 4$, which achieves perfect recall (1.000) at precision 0.275 ($F_2 = 0.655$). This setup ensures no relevant paper is missed, while reducing workload to 90.5% relative to manual review.[2]

## 5 STAGE B – FULL-TEXT SCREENING (FT)

**Problem definition and inputs.** For each candidate $d$ that passes TA, the input is the long document $f_d$ (PDF text with page/section anchors). The model outputs a binary decision $\hat{y}_d \in \{\texttt{include}, \texttt{exclude}\}$ and, when requested, a rationale $\hat{r}_d$ consisting of one or more *verbatim* evidence spans with page/section references plus a brief note. We evaluate on the FT `dev`/`train`/`test` splits (Sec. 3.2). Metrics and validators follow Sec. 3.3; the primary selection metric is $F_2$ and the *Reliability Rate* checks format adherence of rationales (page indices present; spans quoted from $f_d$; note length $\leq$ N tokens).

---

[2]Workload reduction is TN/N, from $N = 1,800$, $n_+ = 47$, recall $= 1.0 \Rightarrow$ TP$= 47$, FN$= 0$. From precision $= 0.275$, FP$\approx 124$, hence TN$= 1,629$, giving WR $= 1629/1800 \approx 0.905$.

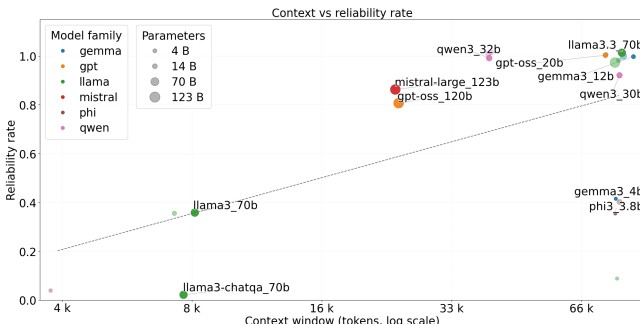

Figure 5: **Model capacity and reliability in FT.** Reliability rate depends jointly on parameter count and context window. Models with sufficient scale and long contexts achieve near-perfect reliability (e.g., gpt-oss-20B, Qwen3-32B, Llama3.3-70B). In contrast, models with large context windows but insufficient parameters (e.g., Gemma3-4B) or with high capacity but limited context (e.g., Llama3-70b) perform poorly. These results motivate a two-step prompting scheme to enforce consistent outputs.

Building on the results from abstract screening, the next set of experiments investigate whether LLMs can extend their capabilities to automate full-text inclusion decisions. This stage introduces distinct challenges: longer contexts, more complex reasoning, and the need for stricter adherence to output formats. We evaluate the same model families under these conditions, focusing on the design factors that govern reliability and performance. In particular, we examine how context length and parameter count interact to determine consistent yes/no outputs, whether Chain-of-Thought (CoT) reasoning and incorporating researcher "thought traces" can guide alignment with nuanced criteria, and how adaptive routing strategies can exploit heterogeneity across models. The overall aim in this stage is to identify the principles that enable dependable full-text screening and to determine the configurations that achieve the strongest results on our benchmark.

Each entry contains the processed full text, a binary inclusion label, and (importantly) quotes and reasoning justifying the label. These rationales not only increase transparency of annotation but also enable conducting experiments that incorporate researcher thought traces into the LLM's prompt. We release the available 2,311 full-text PDFs (subject to license permits), their semi-automated labels and researcher-annotated rationales, as part of SciLitBench.

**Model capacity and reliability.** A key challenge in full-text screening is output reliability: whether the model adheres to the instruction to answer with yes/no only. We define reliability rate as the fraction of completions in the evaluation set conforming to this format. Results reveal that both context window size and parameter count jointly determine reliability (Fig. 5). Models with sufficient capacity achieve near-perfect reliability, while those lacking either depth or context length often fall near 0.4. This motivates the two-step prompting scheme adopted throughout this stage, in which an auxiliary refinement model is explicitly tasked with converting responses into strict yes/no outputs.[3]

**Chain-of-Thought (CoT) prompt.** We compare an unstructured system prompt with a CoT variant that requires the model to first reason through the inclusion criteria before outputting its decision. The CoT approach yields a +5.6% relative gain in $F_2$ (0.559 → 0.591), confirming that structured reasoning can modestly improve alignment with task instructions even in constrained yes/no settings.

**Researcher "thought traces".** We then test whether enriching few-shot examples with researcher rationales (quotes from the text and explanations of inclusion/exclusion decisions) could further guide the models. Incorporating thought traces increases $F_2$ by 28% relative to demonstrations without rationales (0.309 → 0.396; Fig. 3b). This result highlights an important design principle: providing models with examples of not just decisions but the reasoning behind them substantially improves consistency with nuanced inclusion criteria. It suggests that researcher rationales are a powerful way to align LLM behavior with domain-specific review standards.

**Token-aware routing system.** Motivated by the strong dependence of performance on model capacity, we design a token-aware routing system. We first select strong and lenient model pools and partition documents into bins at 2k, 5k, and 10k tokens. For each bin, we fit a logistic regression model to determine optimal ensemble weights. At test time, a document is assigned to the appropriate bin based on its length, and the corresponding weighted voting pool produces the final decision. This adaptive procedure surpasses static majority voting, achieving $F_2 = 0.949$, outperforming the best strong-pool ensemble (0.938), the best lenient-pool ensemble (0.893), and their combined pool

---

[3]To standardize runtime and avoid stack-dependent slowdowns, we cap the context used at inference. Models with native windows ≥128k are evaluated at 80k. Two large models (mistral-large-123B, gpt-oss-120B) are further capped at 24k so that all layers remain on-GPU on our H100-80GB setup, avoiding CPU offload. The longest prompt in our corpus is < 80k tokens, so no example is truncated. These choices trade latency for capacity but do not alter the reliability trends reported.

Figure 6: **Human-AI collaboration workflow** for data extraction guideline construction and refinement.

(0.872). The result establishes routing as an effective mechanism for exploiting heterogeneity across models and inputs, and marks the highest observed performance in our benchmark to date.

# 6 STAGE C – STRUCTURED DATA EXTRACTION

**Problem definition and inputs.** Given an included full text $f_d$ (per TA/FT) and the public schema $\mathcal{S}$, the model must output a JSON object $\hat{z}_d$ that strictly conforms to $\mathcal{S}$ (required keys, types, controlled vocabularies). Typical fields include study design, population descriptors (age range, condition), intervention/comparator descriptors, outcomes, sample sizes, and effect summaries. Validation is performed by the deterministic validator $\mathcal{V}$ from Sec. 3.1. Metrics follow Sec. 3.3: *Exact-match*, *Schema-match* (benign normalization), *Record completeness*, and *Reliability Rate*.

The final set of experiments address the task of data extraction, where the goal is to retrieve specific information of interest from the papers included in the review. In our case, we focus on six fields central to understanding how LLMs are applied to automate reviews: year, domain, review stage, approach, metrics/outcomes, and failure modes/limitations. Unlike inclusion screening, this task is more open-ended, since it is not always possible to predefine the categories that should be assigned to each field (for example, deciding how to scope domains or approaches). To address this, we adopt a collaborative workflow in which one model extracts fine-grained candidate labels while a second model, together with the researcher, co-constructs guidelines that refine and standardize these outputs. Although applied to a limited number of papers, this setup reveals several interesting behaviors and produces high-quality extractions that align with researcher standards (Fig. 6).

In this collaborative researcher–AI environment, we separate the roles of two LLMs: an *extractor* that proposes fine-grained labels for each field, and a *collaborator* that aggregates, revises, and enforces consistency with researcher preferences. This design allows guidelines to emerge iteratively and ensures that extraction outputs are aligned with domain-specific criteria. The extractor first produces candidate labels, which the collaborator then organizes into higher-level categories. The collaborator further generates one guideline per field, which the researcher reviews and edits. At prediction time, the collaborator is tasked with verifying whether extractor outputs conform to the revised guidelines. All prompts, outputs, guidelines, as well as the TODOXXX multi-class labels annotated upon the TODOXXX papers filtered by the full-text screening are released with SciLit-Bench. (Due to space limit, we refer the reader to Appx. D for more details on this task.)

# 7 CONCLUSION

AI and LLMs represent one of the most transformative technologies of our time, with the potential to reshape how scientific discovery is conducted. Literature reviews are a natural starting point: they are central to knowledge synthesis, yet increasingly strained by the growth of the scientific record. We believe SciLitBench establishes a key open dataset dedicated to this challenge, introducing evaluations and design principles that make reliable automation possible.

Our study demonstrates that systematic design choices can substantially improve the reliability of LLMs for literature review automation. Across three key stages, several principles emerge. First, scale matters: larger models with expanded context windows consistently yield more accurate and reliable decisions. Second, prompting strategies are not superficial tweaks but central levers: explicit inclusion and exclusion criteria sharpen model performance, while researcher rationales, expressed as thought traces, provide a powerful mechanism to transfer domain expertise into model behavior, ensuring adherence to nuanced criteria. Third, token-aware routing exploits heterogeneity in both model families and input lengths, surpassing static ensembles and setting new performance benchmarks. Finally, collaborative workflows in data extraction reveal that LLMs can not only propose candidate outputs but also co-develop guidelines with researchers, leading to extractions that preserve rigor while reducing ambiguity.

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

## A SEARCH QUERY

```
(TITLE-ABS-KEY(
"literature review*" OR "systematic review*"
OR "scoping review*" OR "narrative review*"
OR "umbrella review*" OR "rapid review*"
OR "integrative review*" OR "evidence synthesis"
OR "meta-analysis")
AND TITLE-ABS-KEY(
"large language model*" OR "LLM" OR "LLMs" OR "nlp"
OR "natural language processing" OR "transformer*"
OR "AI" OR "artificial intelligence" OR "chatgpt"
OR "GPT-" OR "biogpt" OR "biomedgpt" OR "agrigpt"
OR "spiritualgpt" OR "lifegpt" OR "fingpt" OR "llama"
OR "llama-" OR "medllama" OR "mistral" OR "biomistral"
OR "mixtral" OR "mixtral-" OR "bard" OR "bard-"
OR "bert" OR "bert-" OR "legalbert" OR "rasonbert"
OR "finbert" OR "drbert" OR "biobert" OR "scibert"
OR "clinicalbert" OR "biomedbert" OR "mentalbert"
OR "pubmedbert" OR "claude" OR "claude-" OR "palm"
OR "palm-" OR "gemini" OR "gemini-" OR "gemma"
OR "gemma-" OR "copilot" OR "copilot-" OR "deepseek*"
OR "qwen" OR "qwen-" OR "phi" OR "phi-" OR "sciphi"
OR "scipphi" OR "falcon" OR "falcon-" OR "alpaca"
OR "alpaca-" OR "bloom" OR "bloom-" OR "grok" OR "grok-"))
AND PUBYEAR < 2026
AND NOT PUBDATETXT("July 2025" or "August 2025")
```

## B UNIVERSAL PROMPTS

### B.1 TITLE/ABSTRACT SCREENING

**Basic:**

```
"You are an expert in AI and literature reviews. "
"Given the TITLE and ABSTRACT of a scientific paper: "
"Does this paper use AI to automate any part of the process of a
scientific review paper (of any kind)? "
"Answer with 'yes' or 'no' ONLY"
```

**Inclusion:**

```
"You are an expert in AI and literature reviews. "
"Given the TITLE and ABSTRACT of a scientific paper: "
"Does this paper use AI to automate any part of the process of a
scientific review paper (of any kind)? "
"Include it only if the paper uses AI/ML/NLP/Large Language Models
/LLMs to automate or algorithmically execute a step in evidence
synthesis of scholarly articles (e.g., search, deduplication,
title/abstract screening, full-text screening, data extraction,
risk of bias, study classification, snowballing, meta-analysis). "
"Answer with 'yes' or 'no' ONLY"
```

**Inclusion + Exclusion:**

```
"You are an expert in AI and literature reviews. "
"Given the TITLE and ABSTRACT of a scientific paper: "
"Does this paper use AI to automate any part of the process of a
scientific review paper (of any kind)? "
```

```
"Include it only if the paper uses AI/ML/NLP/Large Language Models
/LLMs to automate or algorithmically execute a step in evidence
synthesis of scholarly articles (e.g., search, deduplication,
title/abstract screening, full-text screening, data extraction,
risk of bias, study classification, snowballing, meta-analysis). "
"Exclude: domain literature reviews not about SR automation;
literature review about the application of AI to a particular
domain; bibliometric tools (e.g., citation-intent) unless
directly used for SR steps. "
"Answer with 'yes' or 'no' ONLY"
```

### B.2  FULL-TEXT SCREENING

**Unstructured:**

```
"You are an expert in AI and literature reviews.
Given the full paper text, carefully analyze the content
and discuss whether the paper uses AI to automate
any part of a scientific review process."
```

**Chain-of-Thought:**

```
"You are an expert in AI and literature reviews. "
"Your task is to assess whether the paper uses AI to
automate any part of a scientific review process. "
"1. First determine if the paper is conducting a
literature review of any kind. "
"2. Second, assess if the paper uses AI or other
NLP methods to automate some aspect of the review. "
"3. Finally, reach a conclusion of whether the paper
is performing a scientific review using AI or not."
```

## C  ADDITIONAL PERFORMANCE ANALYSIS

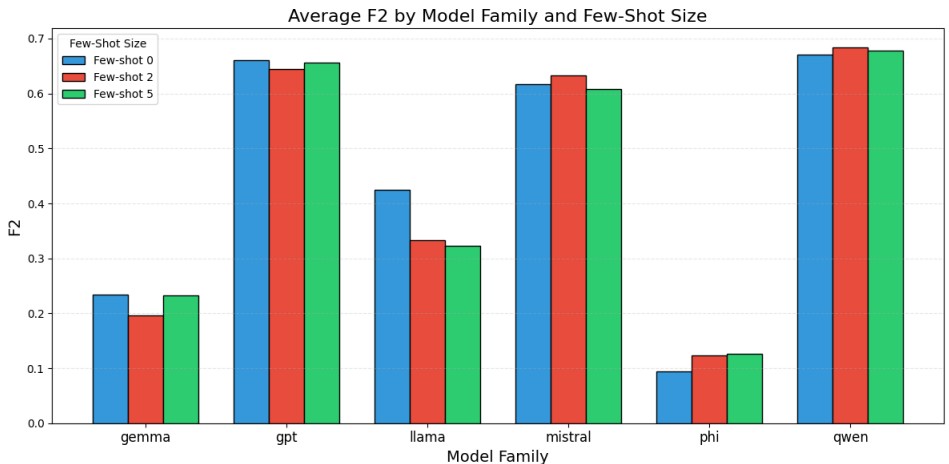

Figure 7: **Effect of few-shot learning on title/abstract screening.**

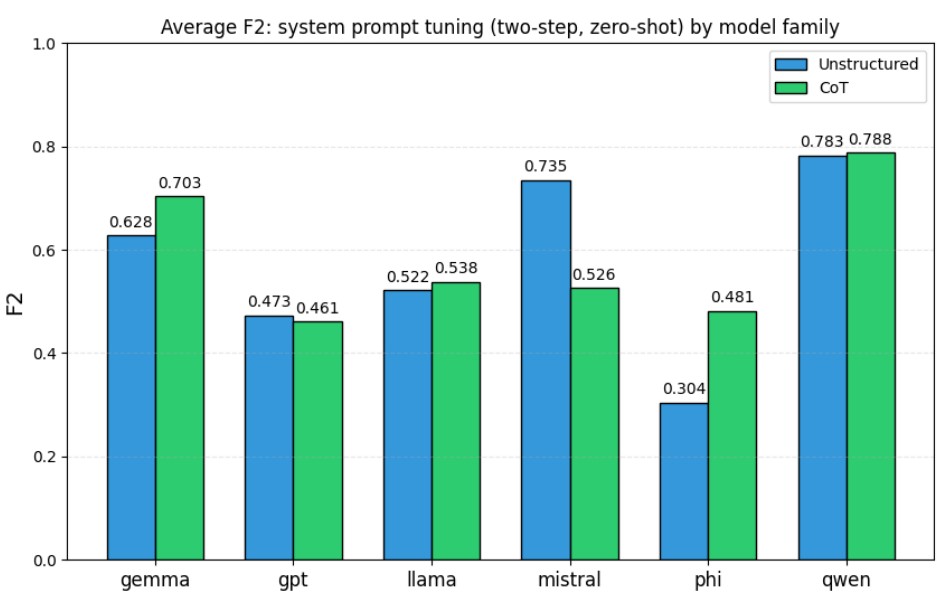

Figure 8: **Effect of chain-of-thought (CoT) prompting on full-text screening.**

# D  STRUCTURED DATA EXTRACTION

## D.1  HUMAN+AI GUIDELINES

A striking observation is the strength of the collaborator's initial draft guidelines. Without explicit instruction, the collaborator produces not only sensible categories but also key steps for the extractor to follow, clarifications for ambiguous cases, decision rules, practical tips, and example mappings between identified instances and proposed categories. These features prove unexpectedly useful. For example, the example–category mappings give the researcher a direct way to validate categorization decisions, while the clarifications and decision rules significantly reduce ambiguity for the extractor.

The researcher then edits the guideline, pruning irrelevant elements (e.g., detailed example mappings), adding "none reported" values where necessary, and sharpening definitions. This step proves crucial: fields such as domain have to be clearly scoped to substantive research areas rather than review methodologies, while metrics requires explicit emphasis on model performance metrics, not study-level statistics. The collaborative revision process thereby transforms a promising but uneven draft into a precise, researcher-approved guideline.

During evaluation, the revised guidelines are provided to the extractor, while the collaborator is tasked with monitoring that the extractor correctly follows them. A schematic overview of this process is shown in Fig. 6 and a full example in Appx. D. Results show a consistent pattern: when extraction is straightforward (e.g., year, domain), the extractor is already correct; when mistakes occur, the collaborator reliably identifies and corrects them. For example, in the review stage field, the extractor identifies multiple stages, but the collaborator correctly filters to only those automated by AI. In metrics/outcomes, the collaborator excludes irrelevant study-level metrics, retaining only those evaluating the automation system. Similarly, in failure modes/limitations, the collaborator ensures that only genuine weaknesses in the review process were kept, not background motivations cited by authors. We encourage readers to consult the supplementary material, which contains excerpts of all model outputs that demonstrate the corrective dynamics between extractor and collaborator LLMs.

This demonstrates a promising design principle for researcher–AI collaboration. Two elements are key: 1) iterative co-construction of the guidelines between collaborator and researcher; 2) role separation between an extractor generating proposals and a collaborator enforcing alignment. Together, these yield a workflow where LLMs not only automate extraction but also learn to adapt to researcher standards, producing outputs that meet human-level standards of sophistication and quality.

