# OpenReview forum: "SciLitBench: Benchmark and Design Principles for LLM-Powered Systematic Literature Reviews"
_ICLR.cc/2026/Conference — Submitted to ICLR 2026_

### Official Review · Reviewer_et5a · 2025-10-31

**Soundness:** 3
**Presentation:** 1
**Contribution:** 2
**Rating:** 2
**Confidence:** 4

**Summary:**

The paper introduces SciLitBench, an open, multi-stage benchmark/protocol for LLM-assisted systematic literature reviews, spanning title/abstract screening, full-text screening, and structured extraction. It curates records and releases frozen splits for TA and FT items with rationales, plus large-scale semi-automated labels for the remainder. Methodologically, it studies prompt design, thought-trace demonstrations, and a token-length–aware routing ensemble, reporting strong F2 improvements and perfect-recall operating points. Figures 2–6 and Tables 2–3 summarize main empirical findings.

**Strengths:**

S1) Clear protocol + frozen splits. Tasks, I/O contracts, and forezen sets for TA/FT are well specificed supporting replication.

S2) Prompting yields good relative gains.

S3) Open-Artifact orientation: Authors emphasize releasing prompts, outputs, scoring scripts, and guidelines to enable "leaderboard-style" comparison.

**Weaknesses:**

W1) Incomplete placeholder stats. The abstract contains “TODOXXX structured data elements” (Abstract, lines 012–020). Similarly, Sec. 1 claims creating “45,291 binary labels and TODOXXX multi-class labels” (Sec. 1, line 086). This undermines reporting completeness.

W2) Semi-automated label prop. risk -- The protocol auto-labels TAs and FTs using the best system with only "sample validation", which may propagate system biases/errors.

W3) "Perfect Recall" comes with *very* low precision. For downstream TA application, the chosen combined voting (t_s=3, t_l=4) achieves recall = 1.0 but precision = 0.275 (Tab 2), implying high FP rate and heavy reviewer load despite the reported 90.5% workload reduction computation.

W4) Limited discussion of training data overlap -- The benchmark topic (AI-for-reviews) likely appears in LLM pretraining corpora. Within pp. 3–6 there’s no audit/guardrail for pretraining overlap or leakage when evaluating open-source LLMs with fixed prompts; this threatens authors' validity claims.

**Questions:**

Q1) Authors should revise this version with significant improvements to their presentation (replace "TODOXXX").

Q2) Data contamination audit -- For the open-source LLMs and frozen splits, can nyou report any steps/heuristics to assess the pretraining overlapwith your benchmark? Or at least a sensitivity check?

Q3) Routing generalization -- The routing ensemble fits length-bin–specific logistic regressions (Tab 3). How robust are the learned weights across topics/length distributions, and what happens if bins or the length cap change? Please add an ablation.

---

### Official Review · Reviewer_w6N2 · 2025-11-01

**Soundness:** 2
**Presentation:** 1
**Contribution:** 2
**Rating:** 2
**Confidence:** 4

**Summary:**

The paper introduces SciLitBench, a benchmark for automating systematic literature reviews with large language models. It evaluates many LLMs on tasks like screening and data extraction, showing that careful prompt design and reasoning examples greatly improve reliability. The study highlights design principles for building trustworthy, efficient AI-assisted review systems.

**Strengths:**

1. The paper conducts extensive experiment with their curated dataset, somewhat bringing insights regarding performances of LLMs in the literature screening task.

2. The experiments concerning scaling law are informative.

**Weaknesses:**

1. This paper has notable presentation problem. For instance, the placeholder span "TODOXXX" appears multiple times, even in abstract; although the authors claim that an example will be presented in Appendix D, it is indeed not there; Figure 7 and 8 are shown without any description or analysis. Therefore, this work leaves me a impression that it still requires more time to be formulated as a publishable manuscript.

2. Some analyses do not fully align with the observation. For instance, in Line 312, the authors claim that "larger models consistently achieve higher F2 scores", while the larger Qwen model results in worse performance compared with its smaller variants. The authors would better provide more detailed interpretation on the results as they are quite a lot of analysis remaining to be made. Similarly, the performance of Phi 3.5 is extremely poor, which requires further descriptions.

3. As a benchmark paper, this study does not provide sufficient details regarding its data, e.g., which "multiple databases" that have been used in the querying, what the genre of literatures is, etc. This drawback considerably undermines the contribution of data curation.

**Questions:**

Please refer to my aforementioned weaknesses.

---

### Official Review · Reviewer_JA7t · 2025-11-02

**Soundness:** 2
**Presentation:** 2
**Contribution:** 3
**Rating:** 2
**Confidence:** 4

**Summary:**

This paper introduces SciLitBench, a large-scale benchmark designed to evaluate the automation of systematic literature reviews. The dataset includes 42,980 abstracts, 2,311 full texts, and structured annotations spanning study metadata, PICO elements, outcomes, and evidence labels. The authors benchmark 22 open-source LLMs under recall-weighted objectives and identify several key factors influencing reliability, including scaling behavior, explicit inclusion/exclusion prompting, and context-length.

**Strengths:**

This paper addresses a valuable problem domain (systematic review automation).

Open the dataset and evaluation rules

Includes long-context evaluations and structured extraction

Attempts to establish practical design principles for reliable LLM screening

**Weaknesses:**

**Major Issues:**
When presenting a new benchmark, it is standard to provide descriptive statistics that give the reader a clear sense of what the dataset covers. Unfortunately, the manuscript and appendix do not provide sufficient detail to understand the main contribution. Even after multiple readings, I still do not have a concrete grasp of the dataset or a clear understanding of how each task was evaluated. Additional clarity is needed.

The paper also lists four contributions; however, not all of them are thoroughly validated. For example, the human-in-the-loop annotation framework is presented as a contribution, but there is no dedicated section in the appendix rigorously evaluating its different versions or comparing it against prior work. This weakens the empirical support for the claimed contribution. Similarly, not really clear what contribution three lists.

**Additional issues**
1. The dataset topic scope is very narrow (AI-for-literature-review papers), reducing generalizability. Ideally, the benchmark would include more topics instead of a **single topic**.

2. Semi-automated label extension using LLMs increases the risk of label propagation bias. What measures are authors taken to mitigate this bias?

3. Similarly, the bias of the annotations can extend to the evaluation. For example, the use of proprietary frontier models to label the majority of the dataset risks circular validation.


4. There is a lack of statistical significance tests or inter-annotator agreement reporting. Please at least consider adding confidence intervals to your results.

5. Statements made in the manuscript implying trustworthy automation are premature, given small validation sets and known LLM hallucination/citation issues.

**Minor Issues:**

1.- There is extensive prior work benchmarking LLMs on systematic review tasks, long-context processing, and structured extraction pipelines. Please expand prior work to include more of these works and contrast how your contribution differs.

2.- This work positions itself as enabling end-to-end systematic review automation, but the benchmark still only simulates isolated filtering/extraction steps rather than a true full-cycle review. This is ok, just make sure to be clearer in the intro and abstract.

3.- Correct abstract (for example, remove TODO) and other mentions of TODOs in the manuscript.

**Questions:**

Please address questions above ^

Also, why are there so many mentions of the applications for systematic reviews in medicine if the single topic covered in this dataset is AI for literature review?

---

### Meta-Review · Area_Chair_22tb · 2026-01-03

**Summary:**

The paper presents a timely, open-source benchmark for automating systematic reviews, distinguished by its focus on long-context processing, structured data extraction, and a clear, reproducible protocol that includes frozen data splits and comprehensive evaluation scripts.

However, there are several major concerns raised by the reviewers:
1. Noticed by multiple reviewers, the manuscript contains multiple "TODO" placeholders in critical areas (including the abstract), lacks essential data statistics, and includes figures (7 and 8) and appendices that are mentioned but entirely missing or unanalyzed.
2. The use of "semi-automated label extension" (using LLMs to label the majority of the dataset) creates a high risk of label propagation bias and circular validation. Furthermore, there is no audit for data leakage, as the topic (AI-for-literature-reviews) is likely present in the models' pre-training data.

There are several concerns regarding the paper, including its narrow scope and lack of generalizability. Additionally, the paper lacks rigorous evaluation and the position of it is weak, for example the paper’s claim to enable “end-to-end” automation is only simulated through isolated steps. To improve its quality, the paper requires significant development and refinement.

**Reviewer Concerns:**

The authors did't respond to the review comments.

**Reviewer Scores:**

n/a

---

### Decision · Program_Chairs · 2026-01-26

Reject